# Comprehensive proteomics and functional annotation of mouse brown adipose tissue

**Jing Li[1]◉, Juan Li[2]◉¤, Wei-Gang Zhao[2]\*, Hai-Dan Sun[1], Zheng-Guang Guo[1], Xiao-Yan Liu[1], Xiao-Yue Tang[1], Zhu-Fang She[3], Tao Yuan[2], Shuai-Nan Liu[3], Quan Liu[3], Yong Fu[2], Wei Sun◉[1]\***

1 Core Facility of Instrument, Institute of Basic Medical Sciences, Chinese Academy of Medical Sciences/ School of Basic Medicine, Peking Union Medical College, Beijing, China, 2 Key Laboratory of Endocrinology of Ministry of Health, Department of Endocrinology, Peking Union Medical College Hospital, Chinese Academy of Medical Science and Peking Union Medical College, Beijing, China, 3 State Key Laboratory of Bioactive Substances and Functions of Natural Medicines, Institute of Materia Medica, Chinese Academy of Medical Sciences and Peking Union Medical College, Diabetes Research Center of Chinese Academy of Medical Sciences, Beijing, China

◉ These authors contributed equally to this work.
¤ Current address: Department of Endocrinology, Beijing Tiantan Hospital, Capital Medical University, Beijing, China
\* sunwei1018@sina.com (WS); xiehezhaoweigang@163.com (W-GZ)

**Data Availability Statement:** Data are publicly available on iProX at https://www.iprox.org/page/PSV023.html;?url=1586916691202OMhj, with the password 9Vs2.

## Abstract

Knowledge about the mouse brown adipose tissue (BAT) proteome can provide a deeper understanding of the function of mammalian BAT. Herein, a comprehensive analysis of interscapular BAT from C57BL/6J female mice was conducted by 2DLC and high-resolution mass spectrometry to construct a comprehensive proteome dataset of mouse BAT proteins. A total of 4949 nonredundant proteins were identified, and 4495 were quantified using the iBAQ method. According to the iBAQ values, the BAT proteome was divided into high-, middle- and low-abundance proteins. The functions of the high-abundance proteins were mainly related to glucose and fatty acid oxidation to produce heat for thermoregulation, while the functions of the middle- and low-abundance proteins were mainly related to protein synthesis and apoptosis, respectively. Additionally, 497 proteins were predicted to have signal peptides using SignalP4 software, and 75 were confirmed in previous studies. This study, for the first time, comprehensively profiled and functionally annotated the BAT proteome. This study will be helpful for future studies focused on biomarker identification and BAT molecular mechanisms.

## 1. Introduction

Two types of adipocytes exist in mammals, white and brown adipocytes [1]. White adipocytes are the major constituents of white adipose tissue (WAT), which contains energy-dense triglycerides that store energy [2]. Brown adipocytes are the major constituents of brown adipose tissue (BAT), which is situated in the interscapular, cervical, mediastinal and retroperitoneal regions [3, 4]. BAT plays a prominent role in thermogenesis and energy expenditure [5]. In

**Funding:** This work was supported by National Basic Research Program of China (No. 2013CB530805, 2014CBA02005), National Key Research and Development Program of China (No. 2016 YFC 1306300,2018YFC0910202), Key Basic Research Program of the Ministry of Science and Technology of China (No. 2013FY114100), National Natural Science Foundation of China (No. 30970650, 31200614, 31400669, 81371515, 81170665, 81560121), Beijing Natural Science Foundation (No. 7173264, 7172076), Beijing cooperative construction project (No.110651103), Beijing Science Program for the Top Young (No.2015000021223TD04), Beijing Normal University (No.11100704), Peking Union Medical College Hospital (No.2016-2.27), CAMS Innovation Fund for Medical Sciences (2017-I2M-1-009) and Biologic Medicine Information Center of China, National Scientific Data Sharing Platform for Population and Health.

**Competing interests:** The authors have declared that no competing interests exist.

**Abbreviations:** 2DLC/MS/MS, two-dimensional liquid chromatography tandem mass spectrometry; 2DLC, two-dimensional liquid chromatography; ACN, acetonitrile; acyl-CoA, acetyl coenzyme A; ADP, adenosine diphosphate; ATP, adenosine triphosphate; BAT, brown adipose tissue; C57BL/6J, inbred mouse strain; cAMP, cyclic adenosine monophosphate; CV, coefficient of variation; EIF2, anti-eIF2 alpha; ErK1 and Erk2, mitogen-activated protein kinase; FABP3, fatty acid-binding protein 3; FASP, filter-aided sample preparation; GO, Gene Ontology; IL-6, interleukin-6; IL-8, interleukin-8; IPA, Ingenuity Pathway Analysis; LCMS, liquid chromatography mass spectrometry; MEK, methyl ethyl ketone; NMN/NAD, nicotinamide mononucleotide; PANTHER, classification: Gene List Analysis; PBEF, pre-B cell colony-enhancing factor; pSTAT3, phosphorylation of signal transducer and activator of transcription 3; RPLC, reversed-phase liquid chromatography; SILAC, Stable Isotope Labeling with Amino Acids in Cell Culture; T2D, type 2 diabetes.

1976, Ricquier et al. found a particular 32,000 Mr membranous protein in rat BAT mitochondria, which was upregulated upon cold exposure [6]. In 1978, the same 32,000 Mr protein was also found by Jean Himms-Hagen [7] and was later named uncoupling protein-1 (UCP-1). UCP-1, found in the inner mitochondrial membrane [8], uncouples the respiratory chain from oxidative phosphorylation, yielding a high oxidation rate and enabling the cellular use of metabolic energy to provide heat [9, 10], indicating its primary role in inducing nonshivering thermogenesis [10]. BAT is a secretory tissue producing batokines, batokines act locally to control the remodelling of BAT, however, and also acts on peripheral organs, such as the liver, skeletal muscle, pancreas, bone, the immune system and the CNS, affecting systemic glucose levels in a thermogenesis-dependent and -independent manner [11].Considering that BAT is closely related to systemic energy metabolism and fat accumulation-related diseases, such as obesity [12], hyperlipidemia, type 2 diabetes (T2D) and cardiovascular disease [13], increasing BAT activity or promoting fat browning may be potential ways to partially prevent or treat these diseases [14, 15].

In exploring the biological functions of BAT, proteomic techniques have proven to be useful [16, 17]. One of the earliest studies on the BAT proteome was led by Sanchez et al. In 2001, these researchers profiled the BAT proteome of C57BL/6 mice using two-dimensional electrophoresis technology. In this study, they separated 87 spots corresponding to 37 protein entries [18]. With the development of mass spectrometry, many efforts have been made to identify more BAT proteins using high-resolution mass spectra in recent years [19–21]. In 2009, Francesca Forner and colleagues compared the brown and white fat mitochondrial proteomes and analyzed the response of the mitochondrial proteome to cold-induced thermogenesis, identifying 2434 proteins in the BAT mitochondria [9]. Recently, Li et al. conducted a BAT proteome analysis using iTRAQ-coupled 2D LC–MS/MS, and 3048 proteins were identified in BAT. By comparing the BAT proteomes from subjects fed a high-fat diet (HFD) and a normal diet (ND), the authors identified 727 differentially expressed proteins [22].

With the development of proteomics technologies [23, 24], the number of identified proteins has quickly increased and even approached that of protein-coding genes in the complete human genome [25]. Compared with the depth of human tissue proteomic research, the mouse BAT proteome has been studied relatively less often. Therefore, in this study, a comprehensive proteomic analysis of BAT profiling was performed using 2D-RPLC and TripleTOF 5600. Qualitative and quantitative analyses were integrated to comprehensively profile the normal BAT proteome. The PANTHER classification system and Ingenuity Pathway Analysis (IPA) software were used to analyze BAT functions and potential biomarkers in BAT. The BAT proteome database developed in this study may facilitate qualitative bioinformatics analysis and provide a basis for future compositional and functional studies focused on BAT.

## 2. Materials and methods

### 2.1. Apparatuses

A TripleTOF 5600 mass spectrometer from AB Sciex (Framingham, MA, USA) and an ACQUITY UPLC system from Waters (Milford, MA, USA) were used.

### 2.2. Reagents

Deionized water from a MilliQ RG ultrapure water system (Millipore, Bedford, MA, USA) was used at all times. LCMS grade acetonitrile (ACN) and formic acid, ammonium bicarbonate, iodoacetamide, dithiothreitol, sequencing-grade modified trypsin, and the protease inhibitor phenylmethanesulfonyl fluoride (PMSF) were purchased from Sigma-Aldrich (St. Louis, MO, USA).

## 3. Brown adipose tissue collection

### 3.1. Animals

All animals were handled according to the Standards for Laboratory Animals (GB14925-2001) and the Guidelines on the Humane Treatment of Laboratory Animals (MOST 2006a) established by the People's Republic of China. The two guidelines were conducted in adherence to the regulations of the Ethics Committee of Peking Union Medical College Hospital Institutional Animal Care and Use Committee (IACUC), and all animal procedures were approved by the Institutional Animal Care and Use Committee (approval number: SCXK Beijing- 2009–0004). All efforts were made to minimize suffering. Six- to eight-week-old female C57BL/6J mice were purchased from HFK Bioscience Laboratories (Beijing, China). According to previous studies diurnal and circadian rhythms related to gene expression patterns and tissue function [26–28]. Female mice were housed at room temperature in a 12:12-h light-dark cycle at 23 ±2˚C, with free access to water and diet. The time of lights on is eight a.m., the time of lights off is eight p.m., and the time of collection tissue is nine a.m. Mice were fed standard chow (SC; 10% lipids) diet for 6–8 weeks. Food and water intake for 24 hours and body weight (once per week) were dynamically monitored any anesthesia method was not used, all mice were decapitated directly.

### 3.2. Preparation of BAT samples

According to previous studies [21, 29, 30] Both gender and age can affect the expression level of BAT protein, so all the mice used in this experiment were C57BL/6 female mice with 6 weeks.

Then washed with a cold saline solution. BATs were snap-frozen in liquid nitrogen and stored at -80 ℃ until analysis. The tissues were cut into small pieces and lysed in buffer solution containing 7 M urea, 2 M thiourea, 65 mM DTE, and 83 mM Tris (Sigma-Aldrich, St. Louis, MO, USA) and then homogenized (IKA R104, Janke& Kunkel KG. IKA-werk, Germany) on ice. The extracts were centrifuged at 20,000 g for 10 min at 4 ˚C, and the supernatant was then stored at -80 ℃. The protein contents of the adipose tissues were determined by the Bradford method with Bradford reagents (Thermo Fischer Scientific, USA).

The six BAT protein samples were pooled into one mixed sample with equal amounts of protein. The mixed sample (100 μg in total) from one group was deoxidized with 20 mM DTT, alkylated with 50 mM IAA and digested with trypsin. After digestion, the peptides were desalted with HLB 3 cc extraction cartridges (Oasis, Waters, Ireland), cleaned with 500 μL of 0.1% formic acid and eluted with 500 μL of 100% ACN. The peptide eluate was vacuum-dried and stored at -80 ˚C. The peptides were processed through columns (micro Bio-spin, nanosep 30k omega; PALL, USA) according to the manufacturer's instructions.

### 3.3. HPLC separation

The lyophilized peptide mixtures were redissolved in 0.1% formic acid and fractionated with a high-pH RPLC column from Waters (4.6 mm × 250 mm, Xbridge C18, 3 μm). The peptide mixture was loaded onto the column in buffer A2 (H2O, pH = 10). The elution gradient, 5%–30% buffer B2 (90% ACN, pH = 10; flow rate, 1 mL/min), was applied over 60 min. The eluted peptides were collected at one fraction per minute. Sixty dried fractions were resuspended in 0.1% formic acid and pooled into 20 samples by combining fractions 1, 21 and 41; 2, 22 and 42; and so on. A total of 20 fractions from one sample were analyzed by LC–MS/MS.

### 3.4. LC–MS/MS analysis

Each sample was analyzed by LC–MS/MS using a reverse-phase C18 self-packed capillary LC column (75 μm×100 mm, 3 μm; Packing: Reprosil-PUR, C18-AQ, 1.9 μm, 120 Å, Dr. Maisch). An elution gradient of 5–30% buffer B1 (ACN and 0.1% formic acid; buffer A1: 98% H2O, 2% ACN, and 0.1% formic acid; flow rate, 0.3 μL/min) was applied over 60 min for the analysis. A TripleTOF 5600 mass spectrometer was used to analyze the peptides eluted from the LC column, and a nanosource was used. Triple TOF 5600 were used to analyze the sample. The MS data were acquired with high sensitivity mode using the following parameters: 30 data-dependent MS/MS scans per every full scan; full scans was acquired at resolution 40,000 and MS/MS scans at 20,000; 35% normalized collision energy, charge state screening (including precursors with +2 to +4 charge state) and dynamic exclusion (exclusion duration 15 s); MS/MS scan range was 100–1800 m/z and scan time was 100 ms.

### 3.5. Data processing

The MS/MS spectra were searched against the SwissProt mouse database from the UniProt website (http://www.UniProt.org) using Mascot software version 2.3.02 (Matrix Science, UK). Trypsin was chosen to calculate the cleavage specificity with a maximum number of allowed missed cleavages of two. Carbamidomethylation (C) was set as a fixed modification. The searches were performed using a peptide and production tolerance of 0.05 Da. A scaffold was used to further filter the database search results using the decoy database method. The following filters were used in this study: 1% false positive rate at the protein level and 2 unique peptides for each protein. After filtering the results with the above filter, the peptide abundances in different reporter ion channels from the MS/MS scan were normalized. The protein abundance ratio was based on the unique peptide results.

### 3.6. Intensity-based absolute quantification (iBAQ) of proteins

Protein abundances were estimated using the iBAQ algorithm [31]. The iBAQ values were calculated by dividing the peptide intensities by the number of theoretically observable peptides from the protein (calculated by in silico protein digestion; all fully tryptic peptides between 6 and 30 amino acids were counted) [32]. The relative iBAQ intensities were computed by dividing the absolute iBAQ intensities from the sum of all absolute iBAQ intensities. The estimated protein abundances were calculated by multiplying the relative iBAQ intensities by the molecular weight of the protein.

### 3.7. Functional annotation

All proteins were assigned a gene symbol using the PANTHER database (http://www.pantherdb.org/). Protein classification was performed based on the functional annotations of the GO project for the cellular compartment, molecular function and biological process categories. When more than one assignment was available, all of the functional annotations were considered in the results. Moreover, all of the selected proteins with significant fold changes were used for pathway analysis using IPA software (Ingenuity Systems Mountain View, CA) for network analysis [33].

### 3.8. SignalP 4.1 Server protein analysis

All proteins were used for adipokine prediction. For this purpose, the SwissProt accession numbers were inserted into SignalP 4.1 (http://www.cbs.dtu.dk/services/SignalP/) [34]. This

software predicts the presence and location of signal peptide cleavage sites in amino acid sequences and suggests proteins with signal peptides.

### 3.9. Post-translational modifications analysis

The MS raw data were processed using the Peaks Online (X build, 1.2.1010.68). High-quality de novo sequence tags were then used by PEAKS DB to search the mouse proteome database. A mass tolerance of 20 ppm and 0.05 Da was set for the precursor ions and fragment ions, respectively. carbamidomethylation was set as fixed modification, Trypsin allowing 3 missed cleavages was chosen as the enzyme. A decoy database was also searched to calculate the false-discovery rate (FDR) using the decoyfusion method.

## 4. Results and discussion

### 4.1. Proteome analysis workflow

To construct a comprehensive dataset for the mouse BAT proteome, proteins were extracted from interscapular BAT harvested from C57BL/6J female mice. The proteins were digested using the filter-aided sample preparation (FASP) method [35]. The pooled digested peptides were separated into 60 fractions by RPLC, mixed into 20 fractions, and then analyzed by nano-RPLC–MS/MS. An iBAQ quantitative analysis method was used to quantify the BAT proteins. The PANTHER classification system and IPA software were used to analyze BAT functions. Brown fat proteins were used for signal peptide prediction via SignalP 4.1 (S1 Fig).

### 4.2. A comprehensive profile of the BAT proteome

**4.2.1. The qualitative identification of BAT proteins.** In this study, BAT samples were used to establish a large database of BAT proteins. As a result, a total of 439,604 spectra, 46,590 unique peptides and 4,949 nonredundant proteins were identified (FDRs < 1% at the peptide and protein level and at least 2 unique peptides for each protein; detailed data are provided in (S1, S2 and S3 Tables). Previous studies reported that brown adipose tissue contained the non-adipocytes including stem cells, [36] immune cells, blood/neural tissue, [37] stromal-vascular (S-V) cells (non-adipocytes) [38], Therefore, our database included the proteins from these non-adipocytes. In addition, blood vessels are also around brown adipose tissue, thus the proteins from plasma were also included in our database. In addition, blood vessels are also around brown adipose tissue, thus the proteins from plasma were also included in our database. In total, 82.8% (4,097/4,949) of the protein groups were identified from the three technical replicates (S1A Fig).

Table 1 summarizes the current studies on the mouse BAT proteome [2, 9, 19–22, 39]. As shown in Fig 1A, compared to the large-scale data, Our study was overlapped 66.8% with Forner et al. [9] and 95% with Li et al. [22]. In addition, we compared the proteins we identified to human adipose tissue proteins, the result showed that 2088 proteins were overlapped with Muller et al. [40]

**4.2.2. The quantitative identification of BAT proteins.** A peak intensity-based semiquantification method (iBAQ) was applied to estimate the abundances of BAT proteins, with 4,495 proteins being quantified. S1C Fig shows the repeatability of the three quantitative experiments, and the correlation between the two protein quantitative intensity runs was approximately 0.98. To reduce the interference caused by technical errors, proteins with coefficients of variation (CVs) greater than 0.3 were excluded (S1B Fig); finally, 4393 proteins were used for further analysis. The iBAQ intensities and estimated protein abundances are provided in S4 Table The dynamic range of the relative abundances spanned five orders of magnitude (Fig 1B).

**Table 1. Proteomic studies focused on the mouse WAT proteome.**

| Year | Mouse strain/age/sex | Number of identifications | Fractionation method | MS instrument | Identification quality (FDR) | Reference |
|------|----------------------|---------------------------|----------------------|---------------|------------------------------|-----------|
| 2001 | 8 week old C57Bl/6J females | 37 | 2-DE | Q-TOF-MS | N/A | Sanchez et al. [18] |
| 2004 | C57BL/6 female mice with 6 weeks | 20 | 2-DE | Q-TOF-MS | N/A | Schmid et al. [39] |
| 2007 | Wistar male rats | 58 | 2-DE | UltraFlex II MALDI-TOF-TOF | N/A | Navet et al. [20] |
| 2009 | C57BL/6 mice (5 to 10 weeks old) | 2434 | SILAC | LTQ-Orbitrap | Protein level< 1% | Forner et al. [9] |
| 2011 | Male and female SLC Sprague-Dawley (SD) rats (5 weeks of age) | 55 | 2-DE | MALDI-TOF-MS | N/A | Choi et al. [21] |
| 2015 | C57BL/6 female mice with 6 weeks | 3048 | iTRAQ | Triple TOF 5600 | Protein level< 1% | Li et al. [22] |
| 2016 | Wild-caught thirteen-lined ground squirrels | 778 | iTRAQ | Velos Orbitrap | Protein level< 1% | Ballinger et al. [19] |
| 2016 | 11 patients | 2019 | TMT | Orbitrap Fusion tribrid | Protein level< 1% | Muller etal. [40] |
| 2019 | C57BL/6 female mice with 6 weeks | 4949 | RPLC | Triple TOF5600 | Protein level<1% | This study |

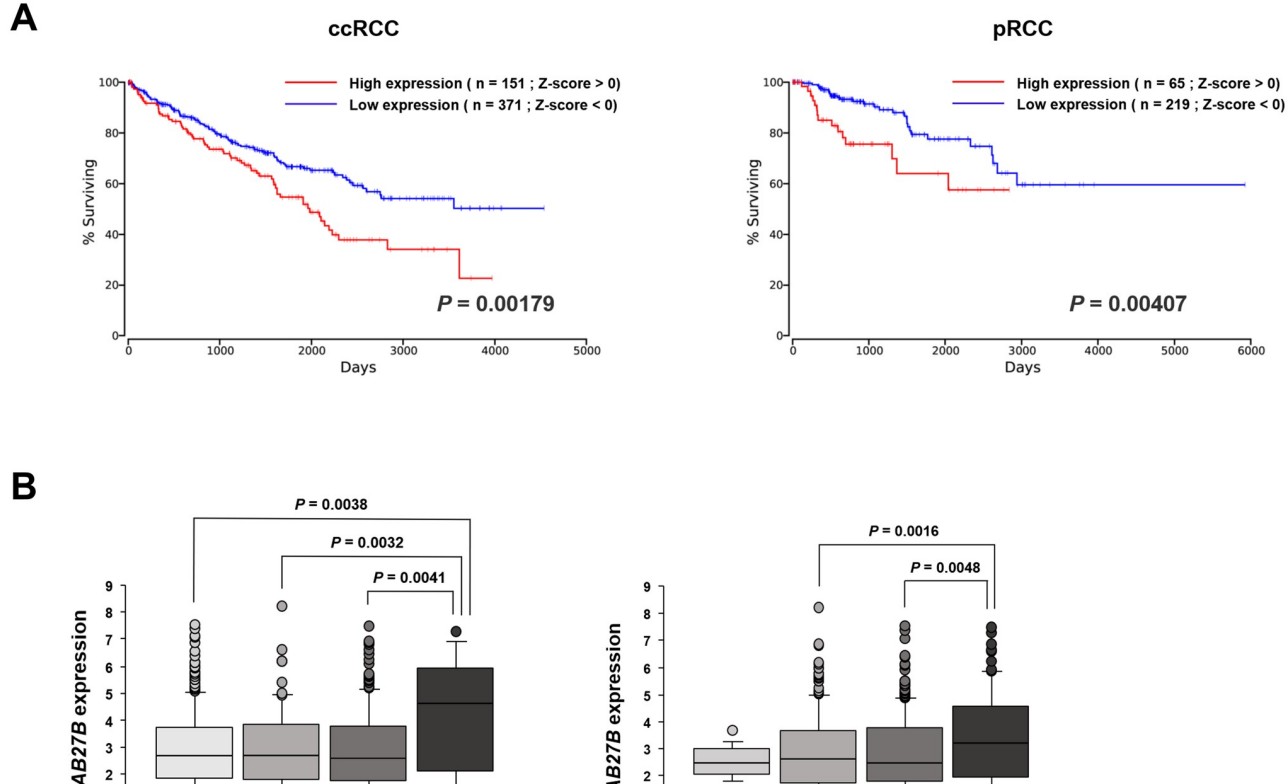

**Fig 1. AT proteome profile analysis. B** (A) A Venn diagram comparing BAT proteome large-scale studies. (B) The quantitative protein abundance ranges in BAT samples and the proteins that included a secreted signal peptide according to the iBAQ algorithm.

The top ten abundant proteins in the BAT proteome are listed in Table 2; these proteins account for 19.5% of the total BAT abundance. Fatty acid-binding proteins, such as FABP3, are essential for cold tolerance and efficient fatty acid oxidation in mouse BAT. FABP3 levels are a determinant of BAT fatty acid oxidation efficiency and represent a potential target for the modulation of energy dissipation [41]. The other proteins included enzymes in the mitochondria that are mainly related to catabolism. Specifically, aconitate hydratase, long-chain specific acyl-CoA dehydrogenase, 3-ketoacyl-CoA thiolase and alpha-enolase were identified as enzymes related to fatty acid oxidation [42, 43], tricarboxylic acid cycle [44] and glycolysis [45]. Electron transfer flavoprotein subunit alpha and cytochrome c oxidase subunit 5A have also been related to mitochondrial respiration [46].

## 4.3. PANTHER classification of the BAT proteome

To analyze the functions of BAT more comprehensively, we divided the BAT proteins into three groups according to their abundance, including high-abundance (abundance: top 90%, 809 proteins), middle-abundance (abundance: 90%-99%, 1,891 proteins) and low-abundance (abundance: 99%-100%, 1,692 proteins) proteins. The functional classification of the proteins identified in BAT was performed using the PANTHER classification system (http://www.pantherdb.org/genes/batchIdSearch.jsp).

From a cellular component perspective (Fig 2A), the GO analysis results showed that more than 50% of BAT proteins were annotated as cell parts and macromolecular complexes, with fewer proteins related to the membrane (10.1%). GO analysis suggested that brown fat proteins were mostly intracellular. Furthermore, among the high-abundance proteins, the proportion of extracellular proteins was higher than that in the middle- and low-abundance proteins.

In terms of molecular function (Fig 2B), the functions of the total proteins included mainly catalytic activity (43.2%) and structural molecule activity (37.6%); and few proteins exhibited transporter activity (5.5%) and signal transduction activity (2.1%). The function of the high-abundance proteins was enriched for catalytic activity (48.8%) and structural molecule activity (30.7%); in contrast, among low- and medium-abundance proteins, 51.1% were related to catalytic activity and 4.4% were related to structural molecule activity. Additionally, a small portion of high-abundance proteins exerted antioxidant activity functions (1.9%). Low-abundance proteins exhibited a higher receptor activity but a lower structural molecule activity than the middle- and high-abundance proteins.

From a biological process point of view (Fig 2C), the metabolic process (25.8%), cellular component organization or biogenesis (10.3%) and cellular process (31.3%) categories were

**Table 2. The top ten abundant proteins in the mouse BAT proteome.**

| Accession | Protein name | iBAQ value (a) | Percentage |
|---|---|---|---|
| P04117 | Fatty acid-binding protein | 4.97E+05 | 3.19% |
| P07724 | Serum albumin | 1.05E+05 | 3.16% |
| P01942 | Hemoglobin subunit alpha | 3.56E+05 | 2.35% |
| P02088 | Cluster of Hemoglobin subunit beta-1 | 3.25E+05 | 2.25% |
| Q99KI0 | Aconitate hydratase | 4.31E+04 | 1.61% |
| P51174 | Long-chain specific acyl-CoA dehydrogenase | 7.02E+04 | 1.47% |
| Q8BWT1 | 3-ketoacyl-CoA thiolase | 8.04E+04 | 1.47% |
| Q99LC5 | Electron transfer flavoprotein subunit alpha | 8.74E+04 | 1.34% |
| P17182 | Alpha-enolase | 6.43E+04 | 1.33% |
| P12787 | Cytochrome c oxidase subunit 5A | 1.83E+05 | 1.29% |

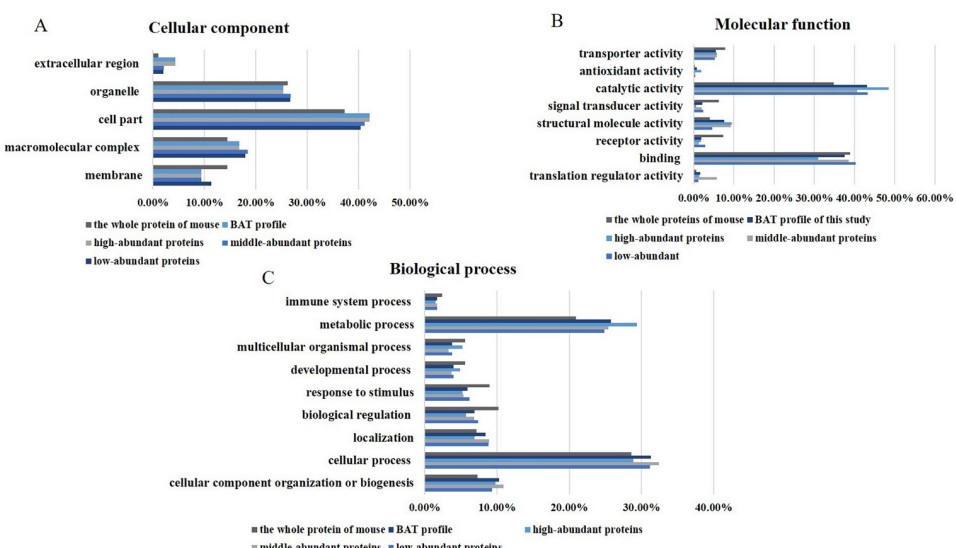

**Fig 2. GO analysis of the BAT proteome.** (A) Cellular component. (B) Molecular function. (C) Biological process.

highly correlated with brown fat proteins. High-abundance proteins were shown to be mainly involved in metabolic processes (29.0%), especially in the generation of precursor metabolites and energy. In contrast, low- and middle-abundance proteins were shown to mainly participate in cellular processes (32.5%), including intracellular signal transduction, cell proliferation and localization processes, such as transport.

## 4.4. Canonical pathway and functional analysis of the BAT proteome

IPA software was used to investigate the functions of the BAT proteome. As shown in Fig 3, BAT functions were mainly related to the regulation of glucose and lipid metabolism, protein synthesis and apoptosis (detailed information is provided in S4 and S5 Tables).

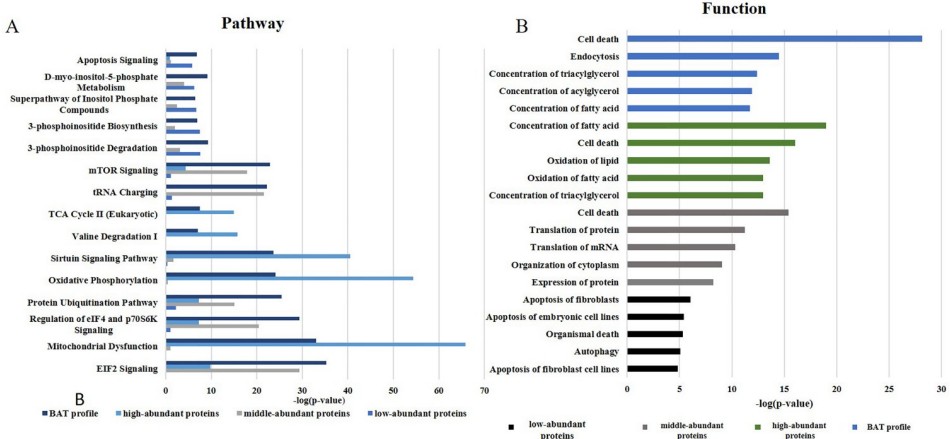

**Fig 3. IPA analysis of the BAT proteome profile.** (A) The top canonical pathways of BAT proteins. (B) Major functions of BAT proteins.

The regulation of glucose and lipoid metabolism mainly involves pathways associated with mitochondrial dysfunction, oxidative phosphorylation, sirtuin signaling and fatty acid β-oxidation [47]. A previous study suggested that BAT could directly improve glucose homeostasis [48–50] and predominantly metabolize lipids for thermogenesis [51]. Protein synthesis focuses on the synthesis of proteins required for organism homeostasis and involves the EIF2 signaling pathway, regulation of eIF4 and the p70S6K signaling pathway and tRNA charging [52]. The apoptosis function was highlighted by apoptosis-related pathways and 3-phosphoinositide degradation, suggesting that apoptosis indeed occurs in BAT under unstimulated conditions; interestingly, cold exposure could inhibit this apoptosis [53].

For high-abundance proteins, the canonical pathways included mitochondrial dysfunction, oxidative phosphorylation, sirtuin signaling, and fatty acid β-oxidation. These metabolic regulatory pathways are mainly performed by oxidizing fatty acids to produce heat [54]. Oxidative phosphorylation releases energy from oxidative sugars, lipids and amino acids. In this process, electrons or protons in the respiratory chain are transferred from ADP phosphorylates to ATP. Fatty acid β-oxidation occurs in the mitochondria. During this process, fatty acids are activated to form acyl-CoA, which then enters the mitochondria and becomes oxidized to produce energy. Carnitine acyl transferase I affects the entry of lipid-CoA into the mitochondria and serves as the rate-limiting enzyme during this oxidation process [55]. More importantly, UCP-1 plays a significant role in this oxidation process. UCP-1 increases the inner mitochondrial membrane conductance for H+ to dissipate the mitochondrial H+ gradient and converts the energy from substrate oxidation into heat [56]. Therefore, the main role of the high-abundance proteins is also reflected in heat generation and energy supply.

For the middle-abundance proteins, the main related pathways were the regulation of eIF4 and p70S6K signaling, the EIF2 signaling pathway and tRNA charging. p70 ribosomal S6 kinase (p70 S6K) has been identified as a potentially important element in the p70S6K signaling pathway for controlling proteolysis in brown adipocytes. In 1992 and 1996, Waldbillig et al. [57] and Moazed et al. [58] proved that insulin increases the phosphorylation of p70S6K, which stimulates protein synthesis in brown adipocytes and inhibits proteolysis, respectively. Moazed et al. found that insulin and its receptor could lead to the activation of PI3-kinase, followed by the phosphoinositide activation of a kinase cascade and p70S6K phosphorylation, which increases proteolysis in brown adipocytes [59]. Increased protein synthesis and mitochondrial biogenesis are expected to promote brown adipocyte proliferation and differentiation, ultimately leading to enhanced BAT growth and development [60].

Low-abundance proteins have been found to be primarily involved in apoptotic signaling pathways, such as 3-phosphoinositide degradation and IL-8 signaling. A previous study performed by Hellman and Hellerström in 1961 suggested that brown fat is associated with apoptosis [61]. The authors showed that apoptosis was an ongoing process in BAT even under normal conditions. Because brown fat is closely related to heat production, reducing brown fat cell apoptosis has become an important aspect of brown fat research. Briscini et al. found that the prolonged exposure of obese animals to cold could prevent brown fat cell apoptosis [62] Lindquist et al. showed that low temperature (4 ˚C) rapidly increased the phosphorylation of the mitogen-activated protein kinases (p42/p44) Erk1 and Erk2 and protected cells in the tissue from apoptosis. The return of mice exposed to cold temperatures to warm temperatures led to a decrease in ErK1 and ErK2 stimulation and an increase in DNA fragmentation and the apoptosis rate [63]. Norepinephrine, which inhibits apoptosis in brown adipocytes, could stimulate the Erk cascade through both a1- and b-adrenergic receptors mediated by $Ca^{2+}$ and cAMP via the Erk kinase MEK. Norepinephrine also stimulates the expression and secretion of basic fibroblast growth factor from cells, which further promotes cell survival via the MEK-

dependent activation of Erk1/2 [63]. The inhibition of apoptosis and retention of more cells promotes an increase in adipocyte cell numbers in the tissue.

## 4.5. Canonical pathway and functional analysis of the BAT mitochondrial proteome

We used GO analysis to annotate the cellular components of all the proteins, and 214 mitochondria proteins were found. These BAT mitochondrial proteins were subdivided into three groups according to their abundance, 72 high-abundance, 90 middle-abundance, 39 low-abundance proteins.

The functions of BAT proteins were mainly related to the regulation of glucose and lipid metabolism, protein synthesis and apoptosis. The mitochondrial proteins were mainly involved in pathways of valine degradation I, isoleucine degradation I, ketogenesis, sirtuin signaling pathway and tRNA charging, which were related to oxidative phosphorylation and amino acid metabolism (Fig 4A).

The high-abundance proteins were mainly involved in amino acid degradation, the middle-abundance proteins were mainly related to amino acid biosynthesis and degradation, and the low-abundance proteins were found to be involved in apoptotic signaling pathways such as death receptor Signaling and thioredoxin Pathway (Fig 4B).

## 4.6. The post-translational modifications of BAT proteins

The MS/MS spectra were processed using Peaks Online (X build, 1.2.1010.68) to identify the post-translational modifications (PTM) of BAT proteome. Total 562 post-translational modifications proteins, 1665 peptides and 4464 MS/MS spectra were found in our data, including 142 acetylation, 117 phosphorylation and 179 methylation etc. (Table 3 and S6 Table). We also quantitatively analyzed the PTM proteins by label-free approach. The PTM proteins abundance accounted for 0.37% of total abundance. Among them methylation showed the highest abundance (0.07%).

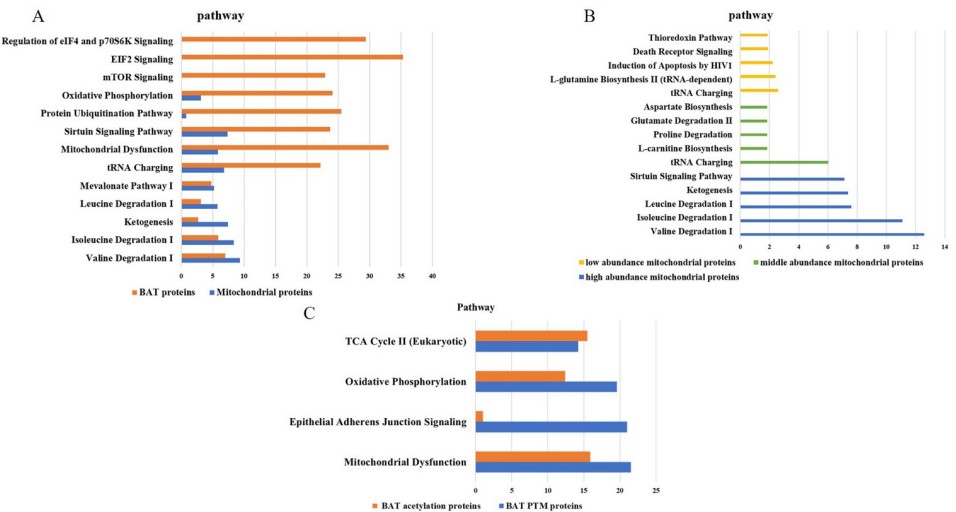

**Fig 4. IPA analysis of the BAT mitochondrial proteome profile and PTM proteins.** (A) The pathways comparisons of BAT proteins and mitochondrial proteins. (B) the pathways of comparisons of high-, middle-, low- abundance proteins of mitochondrial proteins of BAT. (C) The pathway of BAT PTM proteins.

**Table 3. The post-translational modifications of BAT proteins.**

| Name | Protein number | peptide number | #PSM | proportion |
|---|---|---|---|---|
| Acetylation (K) | 142 | 338 | 1070 | 0.07% |
| Amidation | 217 | 302 | 547 | 0.04% |
| Methylation(KR) | 179 | 343 | 901 | 0.09% |
| Phosphorylation (STY) | 117 | 168 | 903 | 0.03% |
| Carboxylation (DKW) | 54 | 83 | 180 | 0.06% |
| Fluorination | 45 | 54 | 76 | 0.00% |
| Formylation (Protein N-term) | 16 | 20 | 47 | 0.00% |
| Formylation | 177 | 357 | 740 | 0.07% |
| sum | | | | 0.37% |

To reveal the function of PTM proteins, they were input IPA for function annotation. The canonical pathways of PTM proteins were involved in mitochondrial dysfunction, epithelial adherens junction signaling and oxidative phosphorylation, which was consistent with BAT function (S3 Fig). Among acetylation proteins, sirtuin 3(SIRT3) was an important protein in BAT function, which is the master deacetylase of deacetylation in the mitochondria. Previous study showed SIRT3 absence in mice resulted in impaired BAT lipid use, whole body thermo-regulation, and respiration in BAT mitochondria without affecting UCP1 expression. Further analysis indicated SIRT3 controlled brown fat thermogenesis by deacetylation regulation of pathways upstream of UCP1 [64].

## 4.7. BAT secretion functions

The ability of BAT to protect against chronic metabolic disease has traditionally been attrib-uted to its capacity to utilize glucose and lipids for thermogenesis [65]. Moreover, BAT also has a secretory role that could contribute to the systemic consequences of its activity. Herein, using SignalP 4.1 software, we predicted 497 proteins with signal peptides, which are potential secretory proteins. Among them, 95, 214, and 188 proteins belonged to the high-, middle-, and low-abundance categories, respectively. The iBAQ method was used to analyze the 497 pre-dicted secreted factors (S7 Table); the dynamic range of the relative abundance spanned five orders of magnitude and accounted for 6.6% of the total BAT abundance. Of these proteins, 72 have been identified as brown fat secretory proteins [66] and 5 have been documented as brown adipokines [65]. For example, a relationship between IL-6 and BAT metabolism has been documented. In rats, IL-6 overexpression was shown to increase thermogenic gene expression and elevate UCP-1 protein levels in BAT, which was mediated by phosphorylation of signal transducer and activator of transcription 3 (pSTAT3) [67, 68]. This result suggests that IL-6 is indeed required to maintain the profound metabolic effects of BAT transplanta-tions and that BAT-derived IL-6 could be a key factor acting as an autocrine or paracrine agent [69].

## 4.8. The discovery of BAT proteome biomarkers

Using IPA analysis, 281 proteins (129 high-abundance proteins, 105 middle-abundance pro-teins, and 48 low-abundance proteins) were found to be related to diseases, and 30 were associ-ated with metabolic disease (S5 Table).

Some of these proteins have been previously reported, such as visfatin. Visfatin was origi-nally isolated as a presumptive cytokine and named a pre-B cell colony-enhancing factor (PBEF) [70]. In 2005, Fukuhara et al. reidentified PBEF as a "new visceral fat-derived

hormone" (visfatin) [71]. Visfatin is an essential NMN/NAD biosynthetic enzyme and is considered a novel adipokine with both intracellular and extracellular enzymatic functions [72]. It stimulates insulin secretion and increases insulin sensitivity and glucose uptake in muscle cells and adipocytes. In 2016, Pisani et al. found that visfatin was highly and preferentially expressed and secreted by human brown and white adipocytes; thus, the authors suggested that visfatin could be considered a rodent brown adipocyte biomarker. However, because available data showed visfatin as a beneficial BAT-derived cytokine for the treatment of obesity and T2D, the search for other potential brown adipocyte-secreted molecules that act in concert with or without visfatin remains urgent [73].

According to previous studies the protein level of brown fat might be changed in different environments such as cold exposure, thermoneutrality, or HFD feeding. [74–76] Therefore, the BAT protein biomarker analysis under above conditions might be important to illustrate the function of brown fat, which should be investigated in the future.

## 5. Conclusion

In this study, we report a large dataset for the mouse BAT proteome based on 2DLC/MS/MS data. The proteins in the comprehensive BAT proteome showed various biological functions and characteristics. This investigation will pave the way for further comparative proteomics approaches focused on the molecular mechanisms involved in BAT disorders.

Because blood contaminant proteins (5.52% abundance) were identified, future studies should aim to remove high-abundance serum proteins. Additionally, more sophisticated instruments (e.g., Orbitrap Fusion Lumos Tribrid mass spectrometer), different quantitative methods (e.g., SILAC labeling) and new techniques (e.g., data-independent acquisition technique) could provide a more precise quantification and higher throughput for BAT studies than those used herein. Furthermore, with the development of genomics, transcriptomics, proteomics and metabolomics, the combination of multiple omics analysis could lead to a better platform for medical biomarker detection, which will allow for the identification of more proteins.

## Supporting information

**S1 Table. List of proteins in the BAT proteome.**
(XLSX)

**S2 Table. List of peptides in the BAT proteome.**
(XLSX)

**S3 Table. Qualitative results for the BAT proteome from the three technical replicates.**
(XLSX)

**S4 Table. The iBAQ intensities and estimated protein abundances for the quantitated BAT proteins.**
(XLSX)

**S5 Table. BAT protein pathways, functions and associated predicted biomarkers.**
(XLS)

**S6 Table. The post-translational modifications of BAT proteins.**
(XLSX)

**S7 Table. iBAQ intensity and estimated protein abundance of potential secretory proteins.**
(XLSX)

**S1 Fig. The workflow of this study.**
(DOCX)

**S2 Fig. Qualitative and quantitative analyses of triplicate LC/MS/MS runs.**
(DOCX)

**S3 Fig. The pathway analysis of BAT proteins and BAT PTM proteins.**
(DOCX)

## Author Contributions

**Conceptualization:** Tao Yuan.

**Data curation:** Zheng-Guang Guo, Wei Sun.

**Formal analysis:** Zhu-Fang She.

**Funding acquisition:** Shuai-Nan Liu.

**Methodology:** Hai-Dan Sun, Yong Fu.

**Resources:** Quan Liu.

**Supervision:** Xiao-Yan Liu, Xiao-Yue Tang.

**Writing – review & editing:** Jing Li, Juan Li, Wei-Gang Zhao.

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
