## [Decision Letter · Decision Letter 0]

24 Oct 2019

PONE-D-19-18696

A comprehensive map and functional annotation of mouse brown adipose tissue

PLOS ONE

Dear Mr sun,

Thank you for submitting your manuscript to PLOS ONE. After careful consideration, we feel that it has merit but does not fully meet PLOS ONE’s publication criteria as it currently stands. Therefore, we invite you to submit a revised version of the manuscript that addresses the points raised during the review process.

Please review and respond to the comments of the reviewers.

We would appreciate receiving your revised manuscript by Dec 02 2019 11:59PM. To enhance the reproducibility of your results, we recommend that if applicable you deposit your laboratory protocols in protocols.io, where a protocol can be assigned its own identifier (DOI) such that it can be cited independently in the future. For instructions see: http://journals.plos.org/plosone/s/submission-guidelines#loc-laboratory-protocols

We look forward to receiving your revised manuscript.

Kind regards,

Jonathan M Peterson, Ph.D.

Academic Editor

PLOS ONE

Journal Requirements:

1.Please complete and submit a copy of the ARRIVE Guidelines checklist, a document that aims to improve experimental reporting and reproducibility of animal studies for purposes of post-publication data analysis and reproducibility: https://www.nc3rs.org.uk/arrive-guidelines. Please include your completed checklist as a Supporting Information file. Note that if your paper is accepted for publication, this checklist will be published as part of your article.

Specifically, please ensure that you revise your methods section to include the following:

•    Please provide details of animal welfare and housing (e.g., shelter, food, water, environmental enrichment).

•    Please specify the method of anaesthesia used.

•    Please specify the method of euthanasia.

•    Please specify the method used for BAT isolation.

2. Thank you for including your ethics statement:  "All animals were handled according to the Standards for Laboratory Animals (GB14925-2001) and the Guidelines on the Humane Treatment of Laboratory Animals (MOST 2006a) established by the People’s Republic of China. The two guidelines adhered to the regulations of the Institutional Animal Care and Use Committee (IACUC), and all animal procedures were approved by the IACUC (approval number: SCXK Beijing- 2009-0004). All efforts were made to minimize suffering. Six- to eight-week-old female C57BL/6J mice were purchased from HFK Bioscience Laboratories (Beijing, China). ".   

To comply with PLOS ONE submissions requirements, please provide the following information in the Methods section of the manuscript and in the “Ethics Statement” field of the submission form (via “Edit Submission”):

a.Please amend your current ethics statement to include the full name of the ethics committee that approved your specific study.

For additional information about PLOS ONE submissions requirements for ethics oversight of animal work, please refer to http://journals.plos.org/plosone/s/submission-guidelines#loc-animal-research  

For additional information about PLOS ONE submissions requirements for animal ethics, please refer to http://journals.plos.org/plosone/s/submission-guidelines#loc-animal-research

4. Please include your tables as part of your main manuscript and remove the individual files. Please note that supplementary tables (should remain/ be uploaded) as separate "supporting information" files

Additional Editor Comments (if provided):

Please review and respond to the comments of the reviewers.

Reviewers' comments:

Reviewer's Responses to Questions

**Comments to the Author**

1. Is the manuscript technically sound, and do the data support the conclusions?

Reviewer #1: Partly

Reviewer #2: Yes

Reviewer #3: Partly

2. Has the statistical analysis been performed appropriately and rigorously? 

Reviewer #1: Yes

Reviewer #2: Yes

Reviewer #3: I don't know

3. Have the authors made all data underlying the findings in their manuscript fully available?

Reviewer #1: Yes

Reviewer #2: Yes

Reviewer #3: Yes

4. Is the manuscript presented in an intelligible fashion and written in standard English?

Reviewer #1: Yes

Reviewer #2: Yes

Reviewer #3: Yes

5. Review Comments to the Author

Reviewer #1: In the entitled “A comprehensive map and functional annotation of mouse brown adipose tissue”, the authors profiled and functionally annotated a large number of proteins in brown adipose tissue using the iBAQ method. The authors further classified them into high-, middle-, and low-abundance proteins by iBAQ values. Additionally, the relationship between various biological functions and their divided levels were discussed. Newly identified BAT proteome could be helpful for future studies to identify markers for obesity treatment or prevention. Thus, their study may be of interest to those in the field. However, there are several important comments that should be addressed by the authors for publication.

Major:

1. Although the current study suggested possible candidates for obesity markers, the authors should provide more information about those proteins. Under cold exposure, thermoneutrality, or HFD feeding, how is the level of the proteins changed?

2. Is there any sex or age difference? According to previous report (Choi, D.K., et al., Cell Physiol Biochem, 2011. 28(5): p. 933-48.), gender-dimorphic protein modulation in BAT may provide conclusive results showing higher expression of numerous proteins involved in thermogenesis and fat oxidation as well as lower expression of proteins contributing to fat synthesis in female rats than in male rats.

3. It is interesting that high, middle, and low abundance proteins are involved in different signaling pathways. However, for signaling transduction, not only the level of proteins but also the post-translational modification of the proteins is important. It will be interesting and important to examine post-translational modifications of the representative signaling proteins in brown adipocytes under different environment.

Reviewer #2: The current study complements and expands upon other related proteomic analyzes of brown adipose tissue. The experiment was conducted in a technically appropriate manner, and the analytics and downstream analyses are appropriate. Some of the techniques are novel. The discussion provides some interesting insight into expression/function. Overall the study is sound, and provides a platform for new useful data that can be utilized in other hypothesis driven experiments focuses on brown adipose tissue function.

The manuscript is well written and easy to follow.

Minor recommendations: Please provide in the methods section the time of day that the tissue was collected in relation to the photoperiod that the mice were maintained on. There are clear differences in gene expression patterns and tissue function (e.g. glucose metabolism) according to the time of day as it related to diurnal and circadian rhythms (e.g. Van der Veen 2012; Zhang et al 2014; Zvonic et al 2006).

van der Veen DR, Shao J, Chapman S, Leevy WM, Duffield GE. A diurnal rhythm in glucose uptake in brown adipose tissue revealed by in vivo PET-FDG imaging. Obesity (Silver Spring). 2012 Jul;20(7):1527-9. doi: 10.1038/oby.2012.78. Epub 2012 Mar 26.

Zvonic S, Ptitsyn AA, Conrad SA et al. Characterization of peripheral

circadian clocks in adipose tissues. Diabetes 2006;55:962–970.

Zhang, N. F. Lahens, H. I. Ballance, M. E. Hughes, and J.

B. Hogenesch, “A circadian gene expression atlas in mammals:

implications for biology and medicine,” Proceedings of the

National Academy of Sciences of the United States of America,

vol. 111, no. 45, pp. 16219–16224, 2014.

Reviewer #3: 

Overview:

Interscapular BAT from BL6/J female mice was used for proteomics analysis using mass spec and 1620 new proteins were reported.  High abundance proteins were mainly related to glucose and fatty acid oxidation, while middle/low abundance proteins were mostly for protein synthesis and apoptosis. 497 proteins were predicted to have signal peptides.  In total the data set is likely useful but in the current presentation lacks detail, clarity, and interpretation that would provide novelty.

Major Concerns:

- title should include protein or proteomics so as to better match the content of the article

- female mice are included in the study, is there reason to anticipate sex differences?

- Ref 11 may be misinterpreted in the Introduction – BAT secretes factors that act in an autocrine/paracrine manner on BAT itself to carry out these functions?? More updated papers on BATokines should be cited, since recent work has investigated BAT secreted factors and their roles in energy metabolism.

- How does the current mass spec data set compare to the set produced in Ref 22 by Li et al? (and any other similar mass spec BAT proteomics studies?) Are there technical considerations (and what do they mean – clarification on Table 1 would help), tissue collection protocols, mouse strain/age/sex that may create different protein data sets?  Fig 2 is helpful but does not include these parameters, and the 3 studies outlined have relatively little overlap with 1316 proteins in common.  Have any studies been done in human WAT or BAT (or beige, the most common brown adipocyte in humans)?

- 6 BAT samples were collected (was overlying WAT carefully removed??) and pooled – how does this affect statistical rigor to have no biological replicates?

- Were some proteins potentially from non-adipocytes? (ie: BAT contains stem cells, immune cells, blood/neural tissue, etc. aside from pre-adipocytes and adipocytes).  In the top 10 proteins, serum albumin, hemoglobins, and potentially other proteins are part of contaminating blood likely... 

- Since numerous mitochondrial proteomic studies in BAT have been undertaken, which of these proteins are thought to be mitochondrial (ie: in mitochondrial function and/or mitochondrially encoded)?

- Is PANTHER protein classification and annotation with GO terms enough for a proteomics data set?  How about pathway analysis in terms of protein interactions and protein networks?

Minor Concerns:

- Fig 1 is a bit vague and not very useful – adding specific details would help the reader orient to the study and analysis of data

- Fig. 3 -4 can not be read – please increase size of Figures and text.  I can’t see any words at all even when I zoom in, so I can not even comment on these data.

6. PLOS authors have the option to publish the peer review history of their article (what does this mean?). If published, this will include your full peer review and any attached files.

Reviewer #1: No

Reviewer #2: No

Reviewer #3: No

---

## [Author Response · Author response to Decision Letter 0]

13 Feb 2020

1. Although the current study suggested possible candidates for obesity markers, the authors should provide more information about those proteins. Under cold exposure, thermoneutrality, or HFD feeding, how is the level of the proteins changed?

Answer: 

Thanks for the reviewer’s good advice. We agree your suggestions that it is important to analyze the BAT proteins under different environment, which is useful to illustrate the function of BAT. 

This study is a pilot work and the main purpose is to profile the mouse brown adipose tissue proteome. And we also try to present the potential BAT function annotation and possible candidates for obesity markers by previous references analysis.

In the future, we will try to analyze the function and the change of these biomarkers under differential environment. We have added above statements in the manuscript (page 16 line 369-372).

And we revised the manuscript as follows:

According to previous studies the protein level of brown fat might be changed in different environments such as cold exposure, thermoneutrality, or HFD feeding.(Peres Valgas da Silva, Hernandez-Saavedra et al. 2019) (Small, Gong et al. 2018) (Kuipers, Held et al. 2019) Therefore, the BAT protein biomarker analysis under above conditions might be important to illustrate the function of brown fat, which should be investigated in the future. 

2. Is there any sex or age difference? According to previous report (Choi, D.K., et al., Cell Physiol Biochem, 2011. 28(5): p. 933-48.), gender-dimorphic protein modulation in BAT may provide conclusive results showing higher expression of numerous proteins involved in thermogenesis and fat oxidation as well as lower expression of proteins contributing to fat synthesis in female rats than in male rats.

Answer:

 Thank you for your suggestions. According to previous studies both sex and age could affect the BAT.

Studies of adult rats and mice indicate that females have a higher threshold for BAT activation (i.e. activate BAT sooner upon cooling than males). In human female BAT has a higher thermogenic capacity than male (Choi, Oh et al. 2011, Harshaw, Culligan et al. 2014)

For age, previous studies reported that both the activity and mass of BAT gradually decreases with the increase of age (Schosserer, Grillari et al. 2018). We have added above references in the manuscript (Line 102-103 on page 5). 

According to previous studies(Choi, Oh et al. 2011, Harshaw, Culligan et al. 2014) (Schosserer, Grillari et al. 2018) Both gender and age can affect the expression level of BAT protein, so all the mice used in this experiment were C57BL/6 female mice with 6 weeks

3. It is interesting that high, middle, and low abundance proteins are involved in different signaling pathways. However, for signaling transduction, not only the level of proteins but also the post-translational modification of the proteins is important. It will be interesting and important to examine post-translational modifications of the representative signaling proteins in brown adipocytes under different environment.

Answer: 

Thanks for the reviewer’s good advice on protein post-translational modifications, therefore, we added “post-translational modifications” section in revised manuscript as followings. 

The MS/MS spectra were processed using Peaks Online (X build, 1.2.1010.68) to identify the post-translational modifications (PTM) of BAT proteome. Total 562 post-translational modifications proteins, 1665 peptides and 4464 MS/MS spectra were found in our data, including 142 acetylation, 117 phosphorylation and 179 methylation etc. (S6 Table). We also quantitatively analyzed the PTM proteins by label-free approach. The PTM proteins abundance accounted for 0.37% of total abundance. Among them methylation showed the highest abundance (0.07%). 

To reveal the function of PTM proteins, they were input IPA for function annotation. The canonical pathways of PTM proteins were involved in mitochondrial dysfunction, epithelial adherens junction signaling and oxidative phosphorylation, which was consistent with BAT function (Fig S3C). Among acetylation proteins, sirtuin 3(SIRT3) was an important protein in BAT function, which is the master deacetylase of deacetylation in the mitochondria. Previous study showed SIRT3 absence in mice resulted in impaired BAT lipid use, whole body thermoregulation, and respiration in BAT mitochondria without affecting UCP1 expression. Further analysis indicated SIRT3 controlled brown fat thermogenesis by deacetylation regulation of pathways upstream of UCP1(Sebaa, Johnson et al. 2019).

This study is a pilot work and the main purpose is to profile the mouse brown adipose tissue proteome, therefore we provided a PTM profile of BAT as you suggested. The representative signaling proteins in brown adipocytes under different environment were an important issue for BAT function analysis, which will be presented in future work.

we revised our study as follows: (Line 320-335 on page 14). 

The MS/MS spectra were processed using Peaks Online (X build, 1.2.1010.68) to identify the post-translational modifications (PTM) of BAT proteome. Total 562 post-translational modifications proteins, 1665 peptides and 4464 MS/MS spectra were found in our data, including 142 acetylation, 117 phosphorylation and 179 methylation etc. (S6 Table). We also quantitatively analyzed the PTM proteins by label-free approach. The PTM proteins abundance accounted for 0.37% of total abundance. Among them methylation showed the highest abundance (0.07%). 

To reveal the function of PTM proteins, they were input IPA for function annotation. The canonical pathways of PTM proteins were involved in mitochondrial dysfunction, epithelial adherens junction signaling and oxidative phosphorylation, which was consistent with BAT function (Fig S3C). Among acetylation proteins, sirtuin 3(SIRT3) was an important protein in BAT function, which is the master deacetylase of deacetylation in the mitochondria. Previous study showed SIRT3 absence in mice resulted in impaired BAT lipid use, whole body thermoregulation, and respiration in BAT mitochondria without affecting UCP1 expression. Further analysis indicated SIRT3 controlled brown fat thermogenesis by deacetylation regulation of pathways upstream of UCP1. (Sebaa, Johnson et al. 2019).

Name Protein number peptide number #PSM proportion

Acetylation (K) 142 338 1070 0.07%

Amidation 217 302 547 0.04%

Methylation(KR) 179 343 901 0.09%

Phosphorylation (STY) 117 168 903 0.03%

Carboxylation (DKW) 54 83 180 0.06%

Fluorination 45 54 76 0.001%

Formylation (Protein N-term) 16 20 47 0.003%

Formylation 177 357 740 0.07%

sum 0.37%

Reviewer #2: 

Minor recommendations: Please provide in the methods section the time of day that the tissue was collected in relation to the photoperiod that the mice were maintained on. There are clear differences in gene expression patterns and tissue function (e.g. glucose metabolism) according to the time of day as it related to diurnal and circadian rhythms (e.g. Van der Veen 2012; Zhang et al 2014; Zvonic et al 2006).

Answer:

Answer:

Thanks for your suggestions. We agree your comments that diurnal and circadian rhythms related to gene expression patterns and tissue function(Zvonic, Ptitsyn et al. 2006)

We have added above information in the manuscript: (Line 95-100 on page 5). 

 “According to previous studies diurnal and circadian rhythms related to gene expression patterns and tissue function(Zvonic, Ptitsyn et al. 2006, van der Veen, Shao et al. 2012, Zhang, Lahens et al. 2014). Female mice were housed at room temperature in a 12:12-h light-dark cycle at 23±2°C ………And the interscapular BATs were excised directly in 9 a.m by tweezers and scissors from the lateral scapular area of the mice back”

Reviewer #3: 

Major Concerns:

- title should include protein or proteomics so as to better match the content of the article

Answer:

Thanks for your suggestions, the title had been revised as below:

“Comprehensive proteomic dataset and functional annotation of mouse brown adipose tissue“

- female mice are included in the study, is there reason to anticipate sex differences?

Answer:

 Thank you for your suggestions. According to previous studies sex could affect the BAT. Studies of adult rats and mice indicate that females have a higher threshold for BAT activation (i.e. activate BAT sooner upon cooling than males). In human female BAT has a higher thermogenic capacity than male (Choi, Oh et al. 2011, Harshaw, Culligan et al. 2014)

We have added above information in the manuscript (Line 102-103 on page 5). 

“According to previous studies(Choi, Oh et al. 2011, Harshaw, Culligan et al. 2014) (Schosserer, Grillari et al. 2018) Both gender and age can affect the expression level of BAT protein, so all the mice used in this experiment were C57BL/6 female mice with 6 weeks.”

- Ref 11 may be misinterpreted in the Introduction – BAT secretes factors that act in an autocrine/paracrine manner on BAT itself to carry out these functions?? More updated papers on BATokines should be cited, since recent work has investigated BAT secreted factors and their roles in energy metabolism.

Answer: Thank you for your comments. According to your suggestion, we have revised the manuscript as followings (Line 48-51 on page 3)

BAT is a secretory tissue, which can produce BATokines. BATokines act locally to control the remodeling of BAT, and they also act on peripheral organs, including the liver, skeletal muscle, pancreas, bone, the immune system and the CNS, thus affect systemic glucose levels in a thermogenesis-dependent and -independent manner.(Klepac, Georgiadi et al. 2019)

1. How does the current mass spec data set compare to the set produced in Ref 22 by Li et al? (and any other similar mass spec BAT proteomics studies?)

Answer: Thanks for your comments.

The protein accession numbers of previous report and our data were transferred to gene entry names using Uniprot website( https://www.uniprot.org/), then we make a qualitative comparison at the gene level.

2. Are there technical considerations (and what do they mean – clarification on Table 1 would help), tissue collection protocols, mouse strain/age/sex that may create different protein data sets? Fig 2 is helpful but does not include these parameters, and the 3 studies outlined have relatively little overlap with 1316 proteins in common. 

Answer: Thank you for your suggestions. 

According to previous studies, the protein identification will be different because of different tissue collection protocols, mouse strain/age/sex (Choi, Oh et al. 2011, Harshaw, Culligan et al. 2014) (Schosserer, Grillari et al. 2018) (Zvonic, Ptitsyn et al. 2006, van der Veen, Shao et al. 2012, Zhang, Lahens et al. 2014) . Besides above factors, many technical factors also influence the results, including protein digestion method, peptide separation method, mass spectrometry and database search software. Therefore, above factors lead to the relatively low overlap between different studies.

To clearly illustrate the results, according to your suggestions, we have added above factor in Table 1.

Have any studies been done in human WAT or BAT (or beige, the most common brown adipocyte in humans)?

Answer: Thank you for your suggestions

In 2016, Muller et al. present comprehensive proteome analysis of human brown and white adipose tissues. 2519 adipocyte proteins were identified (Muller, Balaz et al. 2016) and 2088 were found in our study.

we added the comparison with human BAT in the manuscript (Line 201-202 on page 8).

“In addition, we compared the proteins we identified to human adipose tissue proteins, the result showed that 2088 proteins were overlapped with Muller et al.(Muller, Balaz et al. 2016)

 Year Mouse

strain/age/sex Number of identifications Fractionation method MS instrument Identification quality (FDR) Reference

1 2001 8 week old C57Bl/6J

females 37 2-DE Q-TOF-MS N/A Sanchez et al.(Sanchez, Chiappe et al. 2001)

2 2004 C57BL/6 female mice with 6 weeks 20 2-DE Q-TOF-MS N/A Schmid et al.(Schmid, Converset et al. 2004)

3 2007 Wistar male rats 58 2-DE UltraFlex II MALDI-TOF-TOF N/A Navet et al.(Navet, Mathy et al. 2007) 

4 2009 C57BL/6 mice (5 to 10 weeks old) 2434 SILAC LTQ-Orbitrap Protein level< 1% Forner et al.(Forner, Kumar et al. 2009)

5 2011 Male and female SLC Sprague-Dawley (SD) rats (5 weeks

of age) 55 2-DE MALDI-TOF-MS N/A Choi et al.(Choi, Oh et al. 2011)

6 2015 C57BL/6 female mice with 6 weeks 3048 iTRAQ Triple TOF 5600 Protein level< 1% Li et al.(Li, Zhao et al. 2015)

7 2016 Wild-caught thirteen-lined ground squirrels 778 iTRAQ Velos

Orbitrap Protein level< 1% Ballinger et al.(Ballinger, Hess et al. 2016)

8 2016 11 patients 2019 TMT Orbitrap Fusion tribrid Protein level< 1% Muller etal.(Muller, Balaz et al. 2016)

9 2019 C57BL/6 female mice with 6 weeks 4949 RPLC Triple TOF5600 Protein level<1% This study

- 6 BAT samples were collected (was overlying WAT carefully removed??) and pooled – how does this affect statistical rigor to have no biological replicates?

Answer: Thank you for the reviewers’ suggestions. 

Based on our observations and previous study reports(Vargas-Castillo, Fuentes-Romero et al. 2017), the BAT and WAT can be clearly distinguished by appearance, since BAT has a characteristic yellow-brown color due to its high content of mitochondria and the color of white adipose tissue is white ,thus we took out the BAT directly from the back of the mouse and removed the white fat by color. 

In proteomic database report several samples were usually pooled into one sample to reduce the individual effect on the final results. Then a comprehensive analysis will be done on the pooled sample. This is a normal strategy for proteomic database analysis(Vargas-Castillo, Fuentes-Romero et al. 2017). In this study we also adapted above strategy. 

- Were some proteins potentially from non-adipocytes? (ie: BAT cotains stem cells, immune cells, blood/neural tissue, etc. aside from pre-adipocytes and adipocytes). In the top 10 proteins, serum albumin, hemoglobins, and potentially other proteins are part of contaminating blood likely... 

Answer: Thanks for your question. previous studies reported that brown adipose tissue contained the non-adipocytes including stem cells, (Villarroya, Cereijo et al. 2019)immune cells, blood/neural tissue,(Payab, Goodarzi et al. 2018)stromal-vascular (S-V) cells (non-adipocytes)(Song, Fukui et al. 2010), Therefore, our database included the proteins from these non-adipocytes. In addition, blood vessels are also around brown adipose tissue, thus the proteins from plasma were also included in our database. 

We have added your suggestions in the manuscript: (Line 191-195 on page 8)

 Previous studies reported that brown adipose tissue contained the non-adipocytes including stem cells, (Villarroya, Cereijo et al. 2019)immune cells, blood/neural tissue,(Payab, Goodarzi et al. 2018)stromal-vascular (S-V) cells (non-adipocytes)(Song, Fukui et al. 2010), Therefore, our database included the proteins from these non-adipocytes. In addition, blood vessels are also around brown adipose tissue, thus the proteins from plasma were also included in our database. 

- Since numerous mitochondrial proteomic studies in BAT have been undertaken, which of these proteins are thought to be mitochondrial (ie: in mitochondrial function and/or mitochondrially encoded)?

Answer: Thank you for your suggestions, according to your suggestions, we add the related results about BAT mitochondrial proteins as follows (page 14 line307-319)

We used GO analysis to annotate the cellular components of all the proteins, and 214 mitochondria proteins were found. These BAT mitochondrial proteins were subdivided into three groups according to their abundance, 72 high-abundance, 90 middle-abundance, 39 low-abundance proteins.

The functions of BAT proteins were mainly related to the regulation of glucose and lipid metabolism, protein synthesis and apoptosis. The mitochondrial proteins were mainly involved in pathways of valine degradation I, isoleucine degradation I, ketogenesis, sirtuin signaling pathway and tRNA charging, which were related to oxidative phosphorylation and amino acid metabolism (Fig4A).

The high-abundance proteins were mainly involved in amino acid degradation, the middle-abundance proteins were mainly related to amino acid biosynthesis and degradation, and the low-abundance proteins were found to be involved in apoptotic signaling pathways such as death receptor Signaling and thioredoxin Pathway. (Fig4B)

- Is PANTHER protein classification and annotation with GO terms enough for a proteomics data set? How about pathway analysis in terms of protein interactions and protein networks?

Answer: Thank you for your suggestions

Gene Ontology (GO) project provides the annotation of data. The GO analysis including three parts: cellular component, molecular function, biological process. From these three aspects of annotation, we can learn about the distribution of proteins in cells, the functions and biological processes of our proteins.

But GO analysis only provides an annotation to give us an initial understanding of the proteins in the data, and more accurate analysis of the protein data is mainly through Ingenuity Pathway Analysis (IPA) software.

IPA is based on the database of various biomedical literature and integrated from third-party databases. In addition, IPA use a suite of algorithms and tools for inferring and scoring regulator networks upstream and downstream of gene-expression data based this database. Furthermore, it predicted the related pathway analysis and diseases and function analysis (Kramer, Green et al. 2014). Currently, IPA analysis is a commonly used proteomic analysis. (Wei, Meng et al. 2019). Therefore, we used both GO annotation and IPA analysis in our study. 

Minor Concerns:

- Fig 1 is a bit vague and not very useful – adding specific details would help the reader orient to the study and analysis of data

Answer: Thanks for your suggestion, we adjusted the definition of Fig 1 and moved this figure to the S1 Fig. 

- Fig. 3 -4 can not be read – please increase size of Figures and text. I can’t see any words at all even when I zoom in, so I can not even comment on these data.

 Answer: Thanks for your suggestion, I am sorry about the figure that you cannot read it. We recalibrated the picture definition

---

## [Decision Letter · Decision Letter 1]

11 Mar 2020

PONE-D-19-18696R1

Comprehensive proteomics and functional annotation of mouse brown adipose tissue

PLOS ONE

Dear Mr sun,

Thank you for submitting your manuscript to PLOS ONE. After careful consideration, we feel that it has merit but does not fully meet PLOS ONE’s publication criteria as it currently stands. Therefore, we invite you to submit a revised version of the manuscript that addresses the points raised during the review process.

Minor corrections needed.  Will not need another round of review.

We would appreciate receiving your revised manuscript by Apr 25 2020 11:59PM. To enhance the reproducibility of your results, we recommend that if applicable you deposit your laboratory protocols in protocols.io, where a protocol can be assigned its own identifier (DOI) such that it can be cited independently in the future. For instructions see: http://journals.plos.org/plosone/s/submission-guidelines#loc-laboratory-protocols

We look forward to receiving your revised manuscript.

Kind regards,

Jonathan M Peterson, Ph.D.

Academic Editor

PLOS ONE

Additional Editor Comments (if provided):

Minor corrections needed. Will not need another round of review.

Reviewers' comments:

Reviewer's Responses to Questions

**Comments to the Author**

1. If the authors have adequately addressed your comments raised in a previous round of review and you feel that this manuscript is now acceptable for publication, you may indicate that here to bypass the “Comments to the Author” section, enter your conflict of interest statement in the “Confidential to Editor” section, and submit your "Accept" recommendation.

Reviewer #1: All comments have been addressed

Reviewer #2: (No Response)

2. Is the manuscript technically sound, and do the data support the conclusions?

Reviewer #1: Yes

Reviewer #2: Yes

3. Has the statistical analysis been performed appropriately and rigorously? 

Reviewer #1: Yes

Reviewer #2: Yes

4. Have the authors made all data underlying the findings in their manuscript fully available?

Reviewer #1: Yes

Reviewer #2: Yes

5. Is the manuscript presented in an intelligible fashion and written in standard English?

Reviewer #1: Yes

Reviewer #2: Yes

6. Review Comments to the Author

Reviewer #1: The authors well answered questions we asked, and they revised the manuscript according to the questions.

Reviewer #2: The requested information as stated in the methods section is still insufficient or incorrect. Please state the time of lights OFF and time of lights ON, and time of collection of tissue within the methods section. The authors citation of Van der Veen 2012 (citation 26) is incorrect, please use the correct citation: Van der Veen (2012) A diurnal rhythm in glucose uptake in brown adipose tissue revealed by in vivo PET-FDG imaging. van der Veen DR, Shao J, Chapman S, Leevy WM, Duffield GE. Obesity (Silver Spring). 2012 Jul;20(7):1527-9. doi: 10.1038/oby.2012.78. Epub 2012 Mar 26. PMID: 22447290.

7. PLOS authors have the option to publish the peer review history of their article (what does this mean?). If published, this will include your full peer review and any attached files.

Reviewer #1: No

Reviewer #2: No

---

## [Author Response · Author response to Decision Letter 1]

31 Mar 2020

Thanks for the reviewer’s good advice.

We revised the manuscript as follows:

“Female mice were housed at room temperature in a 12:12-h light-dark cycle at 23±2°C, with free access to water and diet. The time of lights on is eight a.m., the time of lights off is eight p.m., and the time of collection tissue is nine a.m.”

In addition, we have modified reference 26. Thank you very much for your correction.

---

## [Editor Report · Decision Letter 2]

8 Apr 2020

Comprehensive proteomics and functional annotation of mouse brown adipose tissue

PONE-D-19-18696R2

Dear Dr. sun,

We are pleased to inform you that your manuscript has been judged scientifically suitable for publication and will be formally accepted for publication once it complies with all outstanding technical requirements.

With kind regards,

Jonathan M Peterson, Ph.D.

Academic Editor

PLOS ONE

Additional Editor Comments (optional):

all comments addressed. There was only a minor correction needed from previous submission.
---

## [Editor Report · Acceptance letter]

20 Apr 2020

PONE-D-19-18696R2 

Comprehensive proteomics and functional annotation of mouse brown adipose tissue 

Dear Dr. sun:

I am pleased to inform you that your manuscript has been deemed suitable for publication in PLOS ONE. Congratulations! Your manuscript is now with our production department. 

With kind regards,

on behalf of

Dr Jonathan M Peterson 

Academic Editor

PLOS ONE